# Self-Supervised Representations for Multi-View Reinforcement Learning

**Huanhuan Yang**[1]   **Dianxi Shi** [*2,3,1]   **Guojun Xie**[4]   **Yingxuan Peng**[1]   **Yi Zhang**[2]   **Yantai Yang**[3]   **Shaowu Yang**[1]

[1]College of Computer, National University of Defense Technology, Changsha, China
[2]Artificial Intelligence Research Center, Defense Innovation Institute, Beijing, China
[3]Tianjin Artificial Intelligence Innovation Center, Tianjin, China
[4]College of Computer Science and Technology, Nanjing University of Aeronautics and Astronautics, Nanjing, China

## Abstract

Learning policies from raw, pixel images are quite important for the real-world application of deep reinforcement learning (RL). Standard model-free RL algorithms focus on single-view settings and unify the representation learning and policy learning into an end-to-end training process. However, such a learning paradigm is sample-inefficiency and sensitive to hyper-parameters when supervised merely by the reward signals. Based on this, we present Self-Supervised Representations (S2R) for multi-view reinforcement learning, a sample-efficient representation learning method for learning features from high-dimensional images. In S2R, we introduce a representation learning framework and define a novel multi-view auxiliary objective based on the multi-view image states and Conditional Entropy Bottleneck (CEB) principle. We integrate S2R with the deep RL agent to learn robust representations that preserve task-relevant information while discarding task-irrelevant information and find optimal policies that maximize the expected return. Empirically, we demonstrate the effectiveness of S2R in the visual DeepMind Control (DMControl) suite and show its better performance on the default DMControl tasks and their variants by replacing the tasks' default background with a random image or natural video.

## 1 INTRODUCTION

In recent years, deep reinforcement learning (RL) has shown the potential to learn high-quality policies directly from complex environments with high-dimensional states, such as playing Atari video games (Mnih et al., 2015; Hessel et al., 2018) or operating in visual continuous control tasks

(Lillicrap et al., 2016), etc. Note that we can decouple the RL learning process into two sub-processes: representation learning and policy learning. The former aims to abstract features that characterize high-dimensional states, and the latter aims to find optimal policies that maximize the expected cumulative return. However, standard model-free RL algorithms unify these two sub-processes into an end-to-end training procedure, making the learning sample-inefficiency (Lake et al., 2017; Kaiser et al., 2019) when just being supervised by the reward signals. This situation will be aggravated in the real world as collecting interacting data and training specific policies is expensive and time-consuming (Kalashnikov et al., 2018; Akkaya et al., 2019).

Therefore, for RL algorithms, decoupling representation learning and policy learning in one training procedure provides a feasible solution to alleviate the problem of sample inefficiency. Representation learning decomposes high-dimensional data into low-vectored representations that faithfully characterize them (Lesort et al., 2018). Then, policy learning can benefit from these low-dimensional and informative representations, rather than the raw data, to make the task sample efficiently solved. In this paper, we base our method on this idea, first relying on an auxiliary objective to explicitly obtain latent representations, then training the agent upon these representations.

We focus on multi-view RL, which extends RL to multi-view settings. While most RL algorithms solely consider one-view data, multi-view settings release the restrictions that hinder the application of RL to real-life scenarios. Take the smart vehicle as an example, instead of only using one-view data, it fuses multi-view data perceived by multiple sensors to make safe driving decisions. Actually, compared with the paradigm of learning in one-view settings, learning in multi-view settings is more complex due to the increased difficulties of reasoning representations from complicated multiple views. If solved, it can promote the generalization of RL across varying domains, including their applications in the real world. Thus, we propose S2R: **S**elf-**S**upervised **R**epresentations for multi-view reinforcement learning. Our

---

[*]Corresponding author (dxshi@nudt.edu.cn).

*Accepted for the 38[th] Conference on Uncertainty in Artificial Intelligence* (UAI 2022).

key contributions are summarized as follows.

- **Representation learning framework.** To support the representation learning in multi-view RL, we design a specific learning framework. It is composed of the encoder/target encoder network, feature fusion module, view-specific predictor, and multi-view predictor. After learning marginal representations from the encoder network, we use the reparameterization trick to obtain sampled data utilized by the feature fusion module, and further the multi-view predictor to predict self-supervision signals (latent transition function and reward function). Besides, the sampled data are also fed into the view-specific predictor to make predictions.

- **Self-supervision objective.** To learn compressed representations, inspired by the Conditional Entropy Bottleneck (CEB) (Fischer, 2020), we define a new multi-view CEB (MCEB) auxiliary objective. It maximizes the task-relevant information between representations (marginal or joint) and self-supervision signals and compresses away any task-irrelevant information that comes from multi-view image states but is not contained in the self-supervision signals.

- **Representation learning for multi-view RL.** To integrate the representation learning with the multi-view RL training, we incorporate the MCEB objective with the RL objective by optimizing the RL objective on top of the encoder network optimized by the MCEB objective. We follow the common practice (for a given image, data augmentation is used to generate multiple views) in multi-view learning (Bachman et al., 2019; Wang et al., 2021) to produce multi-view data. Empirically, we show that S2R performs better on default visual DMControl tasks (Tassa et al., 2018) and their noisy variants by replacing the tasks' default background with a random image or complex natural video.

## 2 RELATED WORK

**Reconstruction-based representations.** Auto-encoder, an unsupervised learning technique that uses neural networks for representation learning, is the early work that combines with RL in control tasks (Lange and Riedmiller, 2010; Lange et al., 2012; Yarats et al., 2021). These RL agents first trained an encoder via the reconstruction loss, then learned policies based on the representations encoded by the encoder. However, there is no guarantee that the encoder captures useful information for control tasks in practice. Aiming at this problem, researchers proposed to train the encoder jointly with RL dynamics to learn task-oriented and predictive representations (Watter et al., 2015; Wahlström et al., 2015; Hafner et al., 2019, 2020, 2021; Lee et al., 2020a). Although effective, these approaches try to encode all details into embeddings in the reconstruction process of visual images, resulting in the sensibility to task-independent visual changes

and negative effection on performance due to the existence of task-irrelevant information (Zhang et al., 2018).

**Contrastive-based representations.** As a representation learning method, contrastive learning has been widely used in self-supervised settings and made significant progress in the research of image classification and detection (Caron et al., 2020; Xie et al., 2021). It uses data augmentation (Chen et al., 2020) or image patches (Henaff, 2020) to acquire data samples and learns rich representations via similarity functions (Belghazi et al., 2018; Poole et al., 2019) such that the distance between similar pairs is minimized, between dissimilar pairs is maximized. Many works (Kim et al., 2019; Srinivas et al., 2020; Mazoure et al., 2020) have introduced contrastive learning to RL settings to extract predictive features. However, under the effect of contrastive loss, these methods aim to capture all features in the images to maximize the lower bound of the mutual information, making the features containing task-irrelevant information.

**Multi-view and other representations.** To solely extract task-relevant features from high-dimensional data, researchers have tried various methods. Multi-view learning, also known as data fusion or data integration from multiple views data, is an emerging area in machine learning (Zhang et al., 2016). Though abundant in computer vision tasks (Federici et al., 2020; Wang et al., 2019; Wan et al., 2021), it gains less attention on RL decision-making tasks. Chen et al. (2017) proposed the double-task deep Q-Network within multiple views based on double-DQN (Van Hasselt et al., 2016) and dueling-DQN (Wang et al., 2016). Li et al. (2019) defined a framework that generalized partially observable Markov decision processes (POMDPs) to multi-view settings within multiple observation models. In addition, Zhang et al. (2021) introduced the bisimulation metric (Ferns et al., 2011) to learn latent representations that only encode task-relevant information of image observations. Laskin et al. (2020) proposed a plug-and-play module that achieved SOTA performance on the default visual DM-Control tasks by incorporating data augmentations with the RL agent. Lee et al. (2020b) learned compressed representations of the predictive information of RL dynamics through a CEB objective with the CatGen decoder (Fischer, 2020) in the single-view setting. By contrast, our work, S2R, which learns robust representations via an MCEB auxiliary objective, simultaneously takes advantage of the multi-view learning and CEB principle to preserve task-relevant information and ignore task-irrelevant information. We empirically show the performance improvement of S2R against state-of-the-art methods on a variety of visual control benchmarks.

## 3 PRELIMINARIES

**Multi-view Reinforcement Learning.** In this paper, we consider the multi-view reinforcement learning, an extension of RL to multi-view settings, formulated as a Markov

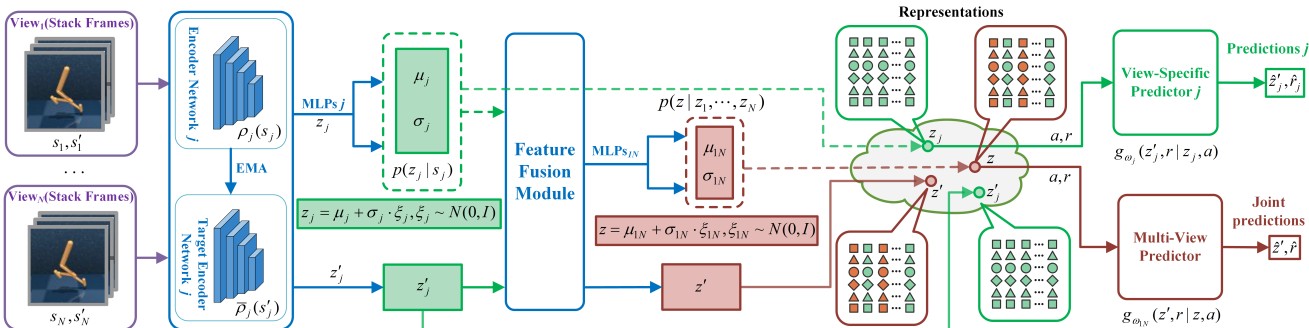

Figure 1: S2R framework. It contains the encoder/target encoder network, feature fusion module, view-specific predictor, and multi-view predictor. Multi-view image state $s_j$ are fed into the encoder network to learn marginal representation $z_j$. Following the reparameterization trick, we obtain sampled representations that successively fed into the feature fusion module and multi-view predictor to predict $z'$ and $r$ and simultaneously into the view-specific predictor to predict $z'_j$ and $r$.

decision process (MDP) $\{S, A, P, r, \gamma\}$. Here, symbols $S$, $A, P(s^{t+1}|s^t, a^t) : S \times A \times S \mapsto [0,1], r(s^t, a^t) : S \times A \mapsto \mathbb{R}$ and $\gamma \in [0,1)$ respectively denote the state space, action space, transition probability of state $s^{t+1}$ when agent takes action $a^t$ at state $s^t$, reward function that maps state $s^t$ and action $a^t$ into real number, and the discount factor. Given $r$ and $\gamma$, the agent aims to learn an optimal policy $\pi$ that maximizes the expected cumulative discounted reward $R = \sum_t \gamma^t r(s^t, a^t)$.

Crucially, we focus on image-based tasks, which means the agent needs to learn policy from pixels. To obtain the multi-view data, referring to the common practice in multi-view learning, we repeatedly apply random data augmentation on the original image state $s^t$ received by the agent to generate diverse sub-images $s^t_j$ as multi-view states, where $j \in [1, N]$ is the view index.

**Soft Actor-Critic.** Soft Actor-Critic (SAC) (Haarnoja et al., 2018) is an off-policy actor-critic algorithm that learns a stochastic policy $\pi_\phi$ to maximize a $\gamma$-discounted and maximum entropy-based return (Ziebart et al., 2008) by optimizing three objectives. Given transition tuples $\tau^t = (s^t, a^t, r^t, s^{t+1})$ sampled from the replay buffer $\mathcal{B}$, the critic minimizes the below Bellman error.

$$L_{Q_{\varphi_i}} = \mathbb{E}_{\tau \sim \mathcal{B}} \left[ \left( Q_{\varphi_i}(s^t, a^t) - (r^t + \gamma V(s^{t+1})) \right)^2 \right] \quad (1)$$

Where $V(s^{t+1})$ is the target value of $s^{t+1}$, defined as:

$$V(s^{t+1}) = \mathbb{E}_{a' \sim \pi} \left( \min_{i=1,2} \bar{Q}_{\bar{\varphi}_i}(s^{t+1}, a') - \alpha \log \pi_\phi(a'|s^{t+1}) \right) \quad (2)$$

Note that SAC maintains two critics ($Q_{\varphi_1}, Q_{\varphi_2}$), two target critics ($\bar{Q}_{\bar{\varphi}_1}, \bar{Q}_{\bar{\varphi}_2}$) and uses the exponential moving average (EMA) to update target network parameters. For the actor, actions are sampled using the reparameterization trick, i.e., $a_\phi(s^t, \xi) = \tan(\mu_\phi(s^t) + \sigma_\phi(s^t) \odot \xi)$ with a standard normalized noise vector $\xi \sim \mathcal{N}(0, I)$, it minimizes:

$$L_{\pi_\phi} = \mathbb{E}_{a \sim \pi} \left[ \alpha \log \pi_\phi(a|s^t) - \min_{i=1,2} Q_{\varphi_i}(s^t, a) \right] \quad (3)$$

For the temperature, given the target entropy $\mathcal{H}$ of the policy distribution, it minimizes:

$$L_\alpha = \mathbb{E}_{a \sim \pi}[-\alpha \log \pi_\phi(a|s^t) - \alpha \mathcal{H}] \quad (4)$$

## 4 S2R FOR MULTI-VIEW RL

To address the learning challenges of multi-view RL mentioned in Sec. 1, we propose S2R, which mainly contains: the representation learning framework, the self-supervision objective, and the combination of S2R with multi-view RL. For readability, we simplify the time index of the transition tuple, replacing $\{s^t, a^t, r^t, s^{t+1}\}$ with $\{s, a, r, s'\}$.

### 4.1 S2R FRAMEWORK

To extract representations from pixel states in multi-view RL, in Fig. 1, we design an S2R representation learning framework. It includes:

(1) Encoder/target encoder network. Both of them are responsible for encoding image states (high-dimensional) into marginal representations (low-dimensional) in a common latent space.

(2) Feature fusion module. Its purpose is to integrate (sampled) marginal representations into joint representations in the common latent space.

(3) View-specific predictor. By inputting the sampled marginal representation together with the action and predicting the latent transition function and reward function, it can maximize task-relevant information and minimize task-irrelevant information in the marginal representation.

(4) Multi-view predictor. By inputting the sampled joint representation together with the action and doing the same prediction, it can effectively extract useful information from the joint representation.

## 4.2 S2R OBJECTIVE

**Two-view CEB.** In 2020, CEB (Fischer, 2020) was proposed. Given the high-dimensional data $X$, it learns representation $Z$ from $X$ to predict label $Y$, defined as $\min_Z \beta I(X; Z|Y) - I(Y; Z)$, expecting that the information captured in $Z$ is maximally relevant to $Y$. In CEB, $I(X; Z|Y)$ is the conditional mutual information, measuring the reduction of uncertainty of $X$ due to learning $Z$ when given $Y$; $I(Y; Z)$ is the mutual information, measuring the reduction of uncertainty of $Y$ due to learning $Z$ (Cover, 1999). Based on CEB, we propose a new MCEB objective to optimize networks related to the S2R framework (Sec. 4.1). For simplicity, we start with a two-view case. Considering the sequential nature of RL, we define $X_1, X_2$ as the current image states, $Z_1, Z_2, Z$ as the current latent representations, and $Y_1, Y_2, Y$ as the rewards and next latent representations. Without loss of generality, we define the two-view CEB objective as:

**obj.**
$$\min_{Z, Z_1, Z_2} \beta_1 I(X_1; Z_1|Y_1) + \beta_2 I(X_2; Z_2|Y_2) - I(Z; Y)$$
$$= \min_{z, z_1, z_2} \beta_1 I(s_1; z_1|z_1', r, a) + \beta_2 I(s_2; z_2|z_2', r, a) -$$
$$I(z; z', r|a)$$
**s.t.** $\quad Z = f_\theta(Z_1, Z_2) \Rightarrow z = f_\theta(z_1, z_2) \quad (5)$

Where $\beta_1, \beta_2$ are regularization factors. To better understand this objective, we show an Information diagram (I-diagram) for $X_1, X_2, Z_1, Z_2, Z, Y_1, Y_2$ and $Y$ in Fig. 2. Intuitively, we observe that: $I(X_1; Z_1) = I(Z_1; Y_1) + I(X_1; Z_1|Y_1)$, $I(X_2; Z_2) = I(Z_2; Y_2) + I(X_2; Z_2|Y_2)$. Thus, to get a minimal and sufficient $Z$, we must minimize redundant information ($I(X_1; Z_1|Y_1)$ and $I(X_2; Z_2|Y_2)$) and maximally preserve relevant information ($I(Z; Y)$, where $Z = f_\theta(Z_1, Z_2)$ is the joint representation of marginal representations $Z_1$ and $Z_2$ fused by the S2R feature fusion module).

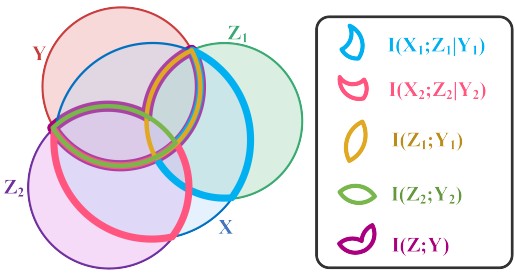

Figure 2: I-diagram of the two-view CEB.

**Optimization of Two-view CEB.** In Eq. (5), it is intractable to directly compute the (conditional) mutual information terms. Fortunately, the variational inference method provides a feasible solution by approximating intractable terms with variational bounds that are easily optimized by standard gradient methods (Kingma and Welling, 2014; Alemi et al., 2017). To get the variational upper bound of Eq. (5),

we first rewrite it below.

$$\min_{Z, Z_1, Z_2} \beta_1(I(X_1; Z_1) - I(Z_1; Y_1)) + \beta_2(I(X_2; Z_2) -$$
$$I(Z_2; Y_2)) - I(Z; Y), \quad Z = f_\theta(Z_1, Z_2)$$
$$= \min_{z, z_1, z_2} \beta_1(I(s_1; z_1) - I(z_1; z_1', r|a)) + \beta_2(I(s_2; z_2) -$$
$$I(z_2; z_2', r|a)) - I(z; z', r|a), \quad z = f_\theta(z_1, z_2) \quad (6)$$

Then, we give the joint probability density function of variables $s_1, s_2, z_1, z_2, z, z_1', z_2', z', r$ and $a$. According to the Bayes's rule, it can be expressed as:

$$p(s_1, s_2, z_1, z_2, z, z_1', z_2', z', r, a) = p(z|s_1, s_2, z_1, z_2, z_1', z_2', z', r, a) \cdot p(z_1|s_1, s_2, z_2, z_1', z_2', z', r, a) \cdot p(z_2|s_1, s_2, z_1', z_2', z', r, a) \cdot p(s_1, s_2, z_1', z_2', z', r, a) \quad (7)$$

Considering $z_1$ is extracted from $s_1$, $z_2$ is extracted from $s_2$, $z$ is fused by $z_1$ and $z_2$, we thus infer that: $z_1$ is independent of variables other than $s_1$, $z_2$ is independent of variables other than $s_2$, and $z$ is independent of variables other than $z_1$ and $z_2$. Therefore, we have:

$$p(s_1, s_2, z_1, z_2, z, z_1', z_2', z', r, a) = p(z|z_1, z_2) \cdot p(z_1|s_1) \cdot p(z_2|s_2) \cdot p(s_1, s_2, z_1', z_2', z', r, a) \quad (8)$$

Based on the standard definition of the (conditional) mutual information, the non-negative property of the Kullback-Leibler divergence (KL-divergence), the above joint probability density function, and the Monte Carlo sampling (Shapiro, 2003), we derive the variational upper bound of Eq. (5) as follows.

$$\beta_1 I(s_1; z_1|z_1', r, a) + \beta_2 I(s_2; z_2|z_2', r, a) - I(z; z', r|a) \leq$$
$$\frac{1}{M} \sum^{M} \Big( \beta_1 [D_{KL}(p(z_1|s_1)||q_1(z_1)) - \mathbb{E}_{z_1 \sim p(z_1|s_1)} \log g_{\omega_1}(z_1', r|z_1, a)] + \beta_2 [D_{KL}(p(z_2|s_2)||q_2(z_2)) -$$
$$\mathbb{E}_{z_2 \sim p(z_2|s_2)} \log g_{\omega_2}(z_2', r|z_2, a)] - \mathbb{E}_{z_1 \sim p(z_1|s_1)}$$
$$\mathbb{E}_{z_2 \sim p(z_2|s_2)} \mathbb{E}_{z \sim p(z|z_1, z_2)} [\log g_{\omega_{12}}(z', r|z, a)] \Big) \quad (9)$$

Where $M$ is the size of data obtained by the Monte Carlo sampling, $g_{\omega_1}(z_1', r|z_1, a)$, $g_{\omega_2}(z_2', r|z_2, a)$ and $g_{\omega_{12}}(z', r|z, a)$ are distributions learned from neural networks (view-specific predictor or multi-view predictor) to approximate real distributions $p(z_1', r|z_1, a)$, $p(z_2', r|z_2, a)$ and $p(z', r|z, a)$, variational distributions $q_1(z_1) \sim N(0, I), q_2(z_2) \sim N(0, I)$ are used to approximate real distributions $p(z_1)$ and $p(z_2)$. Detailed derivations of Eq. (9) are given in Appendix A.

Next, we assume $p(z_1|s_1)$, $p(z_2|s_2)$ and $p(z|z_1, z_2)$ are Gaussian distributions with relative means $(\mu_1, \mu_2, \mu_{12})$ and variances $(\sigma_1, \sigma_2, \sigma_{12})$ learned from MLPs:

$$p(z_1|s_1) = \mathcal{N}(\mu_1(s_1; \psi_1), \sigma_1(s_1; \psi_1))$$
$$p(z_2|s_2) = \mathcal{N}(\mu_2(s_2; \psi_2), \sigma_2(s_2; \psi_2))$$
$$p(z|z_1, z_2) = \mathcal{N}(\mu_{12}(z_1, z_2; \psi_{12}), \sigma_{12}(z_1, z_2; \psi_{12})) \quad (10)$$

In Eq. (10), $\psi_1, \psi_2, \psi_{12}$ are parameters of the MLPs used for learning $p(z_1|s_1)$, $p(z_2|s_2)$ and $p(z|z_1, z_2)$, respectively. To backpropagate the gradient through random variables $z_1$, $z_2$ and $z$, we use the reparameterization trick:

$$\begin{aligned}
z_1 &= \mu_1(s_1; \psi_1) + \sigma_1(s_1; \psi_1) \cdot \xi_1 \\
z_2 &= \mu_2(s_2; \psi_2) + \sigma_2(s_2; \psi_2) \cdot \xi_2 \\
z &= \mu_{12}(z_1, z_2; \psi_{12}) + \sigma_{12}(z_1, z_2; \psi_{12}) \cdot \xi_{12}
\end{aligned} \quad (11)$$

Where $\xi_1 \in \mathcal{N}(0, I), \xi_2 \in \mathcal{N}(0, I), \xi_{12} \in \mathcal{N}(0, I)$ are Gaussian random variables. Therefore, Eq. (9) will be transformed into Eq. (12), the final optimization loss of Eq. (5).

$$\min_{z, z_1, z_2} \frac{1}{M} \sum^M \Big( \beta_1 [D_{KL}(p(z_1|s_1)||q_1(z_1)) - \mathbb{E}_{\xi_1} \log g_{\omega_1}(\\
z_1', r|z_1, a)] + \beta_2 [D_{KL}(p(z_2|s_2)||q_2(z_2)) - \mathbb{E}_{\xi_2} \log g_{\omega_2}(\\
z_2', r|z_2, a)] - \mathbb{E}_{\xi_1} \mathbb{E}_{\xi_2} \mathbb{E}_{\xi_{12}} \log g_{\omega_{12}}(z', r|z, a) \Big) \quad (12)$$

**From Two-view CEB to MCEB.** For cases with more than two views, we can easily generalize the two-view CEB objective to the MCEB objective by adding information terms. Given $N$ views $(X_1, \ldots, X_N)$, it is expressed as:

$$\textbf{obj.} \quad \min_{Z, Z_1, \cdots, Z_N} \sum_{j=1}^N \beta_j I(X_j; Z_j|Y_j) - I(Z; Y) =$$

$$\min_{z, z_1, \cdots, z_N} \sum_{j=1}^N \beta_j (I(s_j; z_j) - I(z_j; z_j', r|a)) - I(z; z', r|a)$$

$$\textbf{s.t.} \quad Z = f_\theta(Z_1, \cdots, Z_N) \Rightarrow z = f_\theta(z_1, \cdots, z_N) \quad (13)$$

Referring to the same derivation process of the two-view CEB objective, the final optimization loss of the MCEB objective (Eq. (13)) can be expressed as follows:

$$\min_{z, z_1, \cdots, z_N} \frac{1}{M} \sum^M \Big( \sum_{j=1}^N \beta_j \Big[ D_{KL}(p(z_j|s_j)||q_j(z_j)) - \mathbb{E}_{\xi_j} \\
\log g_{\omega_j}(z_j', r|z_j, a) \Big] - \mathbb{E}_{\xi_1} \ldots \mathbb{E}_{\xi_N} \mathbb{E}_{\xi_{1N}} \\
\log g_{\omega_{1N}}(z', r|z, a) \Big) \quad (14)$$

### 4.3 INCORPORATE S2R INTO MULTI-VIEW RL

To incorporate S2R into multi-view RL, we simultaneously train the S2R model and the RL agent and treat the S2R loss as an auxiliary loss (Fig. 3). To obtain multi-view image states, we repeatedly apply the random crop augmentation on sampled transition data from the replay buffer and keep it consistent across three consecutive stacked frames to retain the temporal information hidden in the states. This allows the S2R model to infer task dynamics and is more suitable for the RL setting. In Algorithm 1, we give the detailed procedure of integrating S2R with SAC. In our implementation, we use an (target) encoder ($\rho(s_j)/\bar{\rho}(s_j')$), MLPs ($\psi$), view-specific/multi-view predictor ($\omega$) and two views' data. The

first view is not only responsible for the training of the RL agent but also the S2R model together with the second view. For settings with multimodal states (image, text, audio, etc.), we can use $N$ (target) encoders, MLPs, view-specific/multi-view predictors, and the joint latent representation to train the RL agent and S2R model.

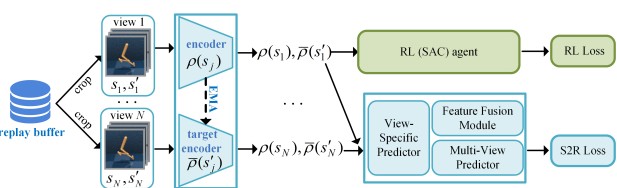

Figure 3: Joint training of the S2R model and RL agent.

---

**Algorithm 1** S2R + SAC pseudo-code

---

1: Initialize: parameters of critic ($\varphi_i, \bar{\varphi}_i$), actor($\phi$), S2R model ($\rho, \bar{\rho}, \theta, \psi, \omega$), temperature ($\alpha$), views $N$, replay buffer $\mathcal{B}$, training step $T$, gradient step $K$, batch size M
2: **for** step $t = 1$ to $T$ **do**
3:     **for** each collection step **do**
4:         Store interaction data: $\mathcal{B} \leftarrow \mathcal{B} \cup (s, a, r, s')$.
5:     **end for**
6:     **for** step $k = 1$ to $K$ **do**
7:         Sample batches $D : \{(s, a, r, s')\}_{m=1}^M$ from $\mathcal{B}$.
8:         Applying data augmentation on $D$, now:
           $D = \{(s_j, a, r, s_j')\}_{m=1}^M, j \in [1, N]$
9:         Compute target value:
           $V = \min \bar{Q}_i(\bar{\rho}(s_1'), a') - \alpha \log \pi(a'|\bar{\rho}(s_1'))$
10:     Update critic:
           $L_{\varphi_i} = [Q_i(\rho(s_1), a) - (r + \gamma V)]^2$
11:     Update actor:
           $L_\phi = \alpha \log \pi(a|\rho(s_1)) - \min Q_i(\rho(s_1), a)$
12:     Update temperature:
           $L_\alpha = -\alpha \log \pi(a|\rho(s_1)) - \alpha \mathcal{H}$
13:     Update S2R model ($\rho(s_j)$, etc.) by $D$, Eq. (14).
14:     Update target critic: $\bar{\varphi}_i = \tau_\varphi \cdot \varphi_i + (1 - \tau_\varphi) \cdot \bar{\varphi}_i$
15:     Update target encoder: $\bar{\rho} = \tau_\rho \cdot \rho + (1 - \tau_\rho) \cdot \bar{\rho}$
16:     **end for**
17: **end for**

---

## 5 EXPERIMENTS

In this paper, we design a variety of experiments to answer the following questions:

- Can S2R have a better sample efficiency in RL visual control tasks (Table 1, Fig. 5 - 8)?

- Is S2R robust to complex settings with the random image distractor or natural video distractor (Fig. 7)?

- Can S2R perform better than existing reconstruction-based, non-reconstruction-based, or contrastive-based RL representation methods (Table 1, Fig. 5 - 7)?

- For S2R, How much information should be preserved for efficient representation? Is it sufficient to merely predict the latent transition function or reward function in MCEB? Is the MCEB objective more suitable than its mutual information or CEB variants? How does S2R perform when the number of views increases? (Fig. 8)

## 5.1 EXPERIMENT SETUP

**DMControl Suite.** To evaluate the performance of S2R, we combine it with the SAC algorithm and focus on visual continuous control tasks in the DMControl Suite (Tassa et al., 2018). Our benchmark includes six different environments under three settings. **(1) Default Setting.** Agent receives pixel states with the default background. **(2) Image Distractor Setting.** Agent receives pixel states with the random image as the background. **(3) Natural Video Setting.** Agent receives pixel states with the natural video selected from the "arranging flowers" class of the Kinetics dataset (Kay et al., 2017) as the background. In Fig. 4, We show snapshots of pixel states in the above settings.

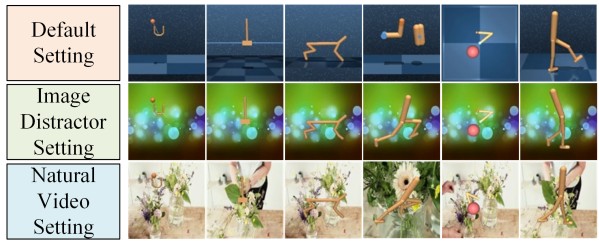

Figure 4: Tasks from left to right are ball-in-cup catch, cartpole swingup, cheetah run, finger spin (the first row)/walker run (the second/third row), reacher easy, and walker walk.

**Implementation.** We base our S2R method on the implementation of RAD (Laskin et al., 2020) [1] and use most of its default parameters, including the learning rate, action repeat, etc. Specially, we use a desktop with an 8-core CPU, and two Nvidia GeForce RTX 3090 for each benchmarking. In our experiments, figures show the mean and standard error across five seeds unless specified otherwise. Besides, we use random crop augmentation on the agent's $100 \times 100$ original image states to obtain $84 \times 84$ multi-view states. Full implementation details and hyper-parameters are listed in Appendix B.

## 5.2 BASELINE ALGORITHM

In this paper, we compare S2R + SAC with some state-of-the-art pixel-based RL methods. DBC (Zhang et al., 2021) learns effective representations for downstream control tasks through the bisimulation metric. RAD (Laskin et al., 2020) uses augmented data to train policy. CURL (Srinivas et al.,

[1] https://github.com/MishaLaskin/rad

2020) combines contrastive learning objective with model-free RL agent. SLAC (Lee et al., 2020a) learns stochastic sequential models via a variational inference objective. PlaNet (Hafner et al., 2019) and Dreamer (Hafner et al., 2020) are two model-based algorithms, they both learn a world model and respectively choose actions via online planning and long-horizon imagination. SAC + AE (Yarats et al., 2021) combines auto-encoder with model-free RL algorithm via an auxiliary reconstruction loss. Pixel SAC is the SAC (Haarnoja et al., 2018) algorithm with image inputs, while State SAC operates on proprioceptive states (positions, velocities, etc.). Besides, in DBC, we use the same action repeat as RAD and S2R to make a fair comparison.

## 5.3 MAIN RESULTS

**Default Setting Results.** To evaluate the sample efficiency of our method, we first give the median scores achieved by S2R + SAC along with the baselines at DMControl100k (low sample performance) and DMControl500k (asymptotical optimal performance) benchmarks [2] in Fig. 5 and show their relative scores on 6 control tasks in Table 1 and Fig. 6. In Fig. 5, S2R + SAC achieves 1.14x/1.04x higher median scores than State SAC, 1.59x/1.05x higher median scores than CURL, and 6.69x/5.12x higher median scores than Pixel SAC at 100k/500k environment steps, showing that S2R + SAC has a higher sample efficiency. In Table 1, S2R + SAC, which integrates MCEB-based representation learning with model-free RL learning, is the state-of-the-art algorithm on all (6 out of 6) visual DMControl tasks on both DMControl100k and DMControl500k benchmarks. It achieves impressive results, exceeds the performance of best-performing RAD and CURL, matches the performance of State SAC operating from proprioceptive states, and significantly improves the performance of Pixel SAC on both DMControl100k and DMControl500k benchmarks. In Fig. 6, the learning curves of S2R + SAC and DBC again confirm the better sample efficiency of S2R + SAC.

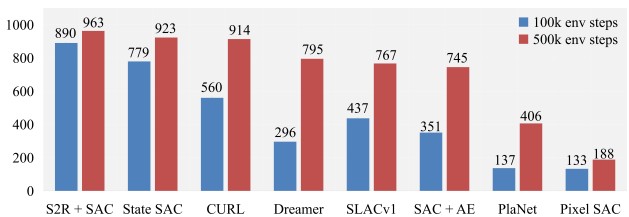

Figure 5: Performance of S2R + SAC relative to baselines averaged across 10 seeds in the default setting. Results are the medians of 6 pixel-based control tasks in Table 1, and data other than S2R + SAC is reported in CURL.

[2] DMControl100k/DMControl500k refers to 100k/500k environment or simulator steps, which is equal to 50k/250k policy steps if the action repeat is set to 2.

Table 1: We report scores (mean and standard deviation) for S2R + SAC and baselines (report by RAD) on DMControl500k and DMControl100k. Results are statistics by averaging the scores of 10 seeds on 6 control tasks. In both benchmarks, compared with existing baselines, S2R + SAC achieves state-of-the-art performance on all (6 out of 6) control tasks.

| 500K STEP SCORES | S2R + SAC | RAD | CURL | PlaNet | Dreamer | SAC + AE | SLACv1 | PIXEL SAC | STATE SAC |
|---|---|---|---|---|---|---|---|---|---|
| FINGER, SPIN | **983** | 947 | 926 | 561 | 796 | 884 | 673 | 192 | 923 |
|  | ± 5 | ±101 | ±45 | ±284 | ±183 | ±128 | ±92 | ±166 | ±211 |
| CARTPOLE, SWING | **869** | 863 | 845 | 475 | 762 | 735 | - | 419 | 848 |
|  | ± 10 | ±9 | ±45 | ±71 | ±27 | ±63 |  | ±40 | ±15 |
| REACHER, EASY | **981** | 955 | 929 | 210 | 793 | 627 | - | 145 | 923 |
|  | ±5 | ±71 | ±44 | ±44 | ±164 | ±58 |  | ±30 | ±24 |
| CHEETAH, RUN | **837** | 728 | 518 | 305 | 570 | 550 | 640 | 197 | 795 |
|  | ± 21 | ±71 | ±28 | ±131 | ±253 | ±34 | ±19 | ±15 | ±30 |
| WALKER, WALK | **950** | 918 | 902 | 351 | 897 | 847 | 842 | 42 | 948 |
|  | ±19 | ±16 | ±43 | ±58 | ±49 | ±48 | ±51 | ±12 | ±54 |
| CUP, CATCH | **978** | 974 | 959 | 460 | 879 | 794 | 852 | 312 | 974 |
|  | ±5 | ±12 | ±27 | ±380 | ±87 | ±58 | ±71 | ±63 | ±33 |
| 100K STEP SCORES | S2R + SAC | RAD | CURL | PlaNet | Dreamer | SAC + AE | SLACv1 | PIXEL SAC | STATE SAC |
| FINGER, SPIN | **876** | 856 | 767 | 136 | 341 | 740 | 693 | 224 | 811 |
|  | ±43 | ±73 | ±56 | ±216 | ±70 | ±64 | ±141 | ±101 | ±46 |
| CARTPOLE, SWING | **868** | 828 | 582 | 297 | 326 | 311 | - | 200 | 835 |
|  | ±9 | ±27 | ±146 | ±39 | ±27 | ±11 |  | ±72 | ±22 |
| REACHER, EASY | **961** | 826 | 538 | 20 | 314 | 274 | - | 136 | 746 |
|  | ±40 | ±219 | ±233 | ±50 | ±155 | ±14 |  | ±15 | ±25 |
| CHEETAH, RUN | **605** | 447 | 299 | 138 | 235 | 267 | 319 | 130 | 616 |
|  | ±22 | ±88 | ±48 | ±88 | ±137 | ±24 | ±56 | ±12 | ±18 |
| WALKER, WALK | **897** | 504 | 403 | 224 | 277 | 394 | 361 | 127 | 891 |
|  | ±42 | ±191 | ±24 | ±48 | ±12 | ±22 | ±73 | ±24 | ±82 |
| CUP, CATCH | **968** | 840 | 769 | 0 | 246 | 391 | 512 | 97 | 746 |
|  | ±6 | ±179 | ±43 | ±0 | ±174 | ±82 | ±110 | ±27 | ±91 |

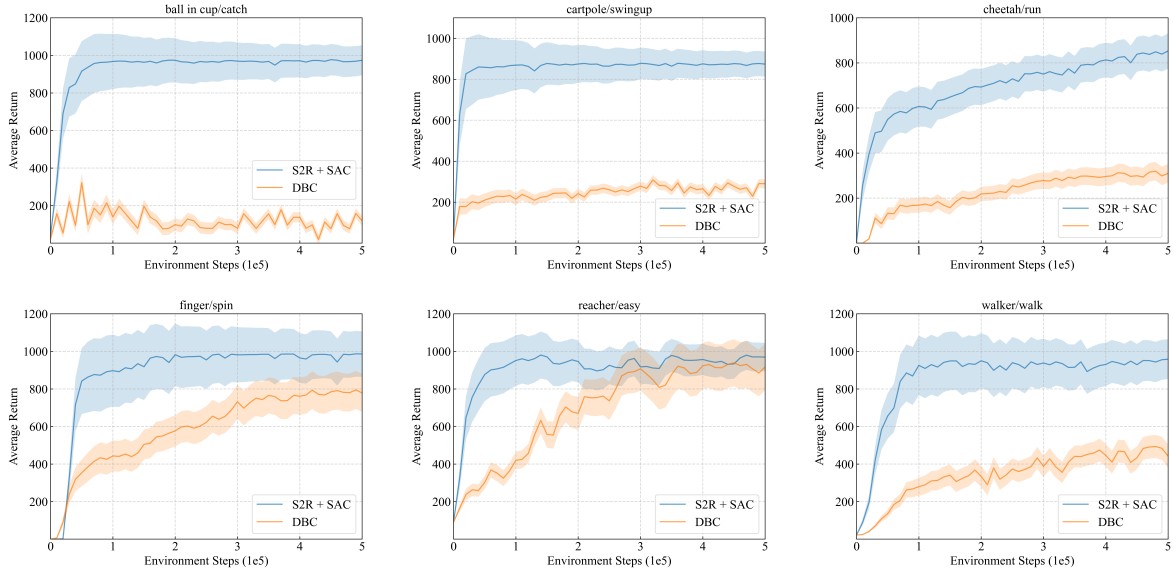

Figure 6: Learning curves in the default setting, a supplement to Table 1. We benchmark S2R + SAC with DBC. Results show that S2R + SAC outperforms DBC and achieves impressive performance on all 6 control tasks.

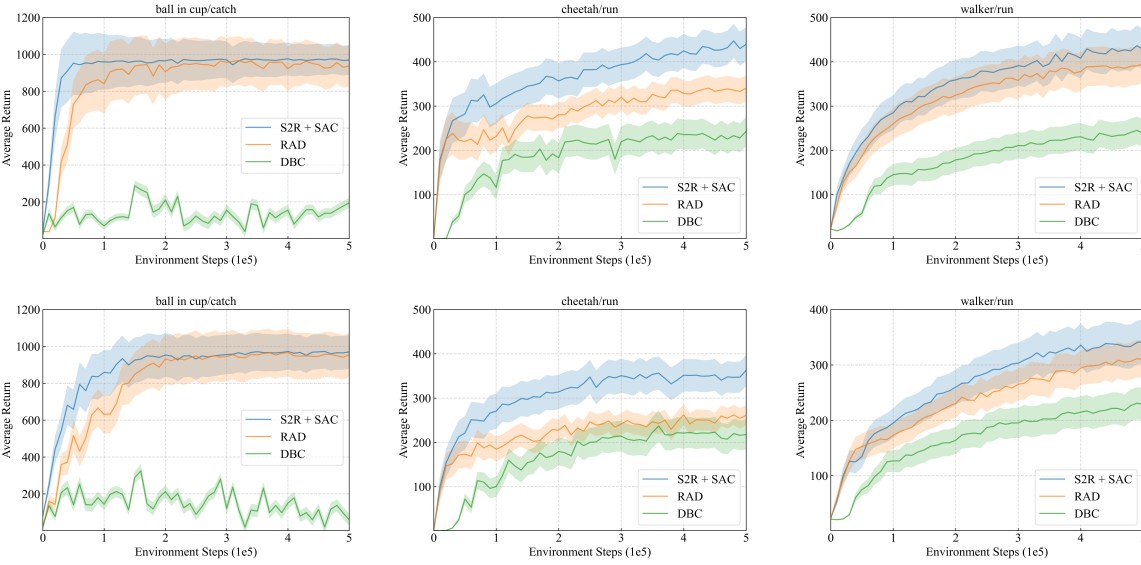

Figure 7: Performance of S2R + SAC. **Top row**: Results in the image distractor setting. **Last row**: Results in the natural video setting. We benchmark S2R + SAC with RAD and DBC in both settings, and results confirm the better performance of S2R + SAC. Additional results can be found in Appendix C.

**Image Distractor Setting Results.** Then, we evaluate S2R performance in the image distractor setting by replacing the tasks' background with a random image. In the top row of Fig. 7, we give the results of three tasks (ball-in-cup catch, cheetah run, and walker run). Results show that S2R + SAC performs comparably or better than RAD, and substantially outperforms DBC, proving that S2R can discard task-irrelevant information when learning representations.

**Natural Video Setting Results.** Next, we evaluate S2R + SAC, RAD, and DBC in a more complex setting by introducing the natural video as the background. In the last row of Fig. 7, we give the results of three tasks (ball-in-cup catch, cheetah run, and walker run). We notice that compared with RAD and DBC, S2R + SAC again performs better and has a higher sample efficiency. This attributes to our well designing of S2R, which makes the agent only focus on task-related features, insensitive to task-irrelevant visual changes, and thus providing robust representations for the training of the actor and critic.

**Ablation Studies.** Finally, in the cheetah run task in Fig. 8, we investigate how S2R is affected by the regularization factors, predictive data ($Y_1$, $Y_2$ and $Y$), optimization objectives, and the number of views. **(1) MCEB regularization factors.** In the MCEB objective, regularization factors are related to the trade-off between the sufficiency and robustness of the representation, and we use an exponential scheduler in all experiments. As seen from Fig. 8(a), in MCEB, too-high values block information essential to the predictive data, while too-small values reduce the benefit of regularization. Results prove the rationality of the set values of the regularization factors in MCEB. **(2) MCEB predictive data.** To utilize the

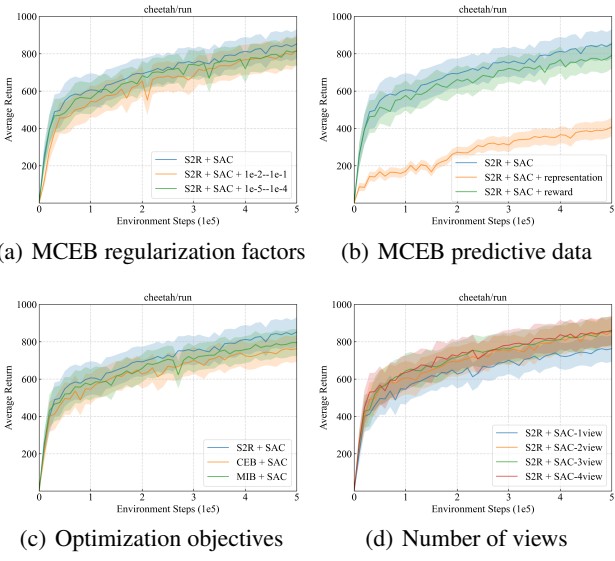

(a) MCEB regularization factors    (b) MCEB predictive data

(c) Optimization objectives    (d) Number of views

Figure 8: Results in the default setting for ablation studies. (a) compares MCEB regularization factors, (b) compares MCEB predictive data, (c) compares MCEB optimization objectives, and (d) compares the number of views $N$ in MCEB. Additional results can be found in Appendix C.

sequential nature of RL, the predictive data in the MCEB objective can be the reward and next latent representation or either of them. However, our experiment results in Fig. 8(b) show that simultaneously predicting the latent transition function and reward function is better than predicting either of them alone. **(3) MCEB optimization objectives.** With a

slight modification to the MCEB objective, its two variants that are similar to reported works can be obtained. The first variant is equal to PI-SAC (Lee et al., 2020b), which optimizes the representation model by the CEB principle in the single-view RL setting. The second variant is equal to MIB (Multi-view Information Bottleneck) (Wang et al., 2019), which replaces the CEB term $I(X_j; Z_j | Y_j)$ with the IB term $I(X_j; Z_j)$ in MCEB. Compared to these two variants, our results in Fig. 8(c) show the better performance and higher sample efficiency of the MCEB objective, confirming the necessity of including multiple views and using the CEB principle in S2R. **(4) Number of views in MCEB.** We further ablate the number of views $N$ included in the MCEB objective to understand its effect on the S2R performance. As we can see from Fig. 8(d), the MCEB objective can benefit from multi-view data (especially when it contains the complementary information) to learn robust representations that improve performance, whereas this is premised on the increase of the training time (as the increase of $N$ means a larger computational demand). To strike a balance between the training time and the method performance, we choose to set the number of views to 2.

## 6 DISCUSSION

In this paper, we present S2R, a multi-view self-supervised representation learning method to learn efficient and sufficient representations for the policy learning of the RL agent based on the multi-view data and CEB principle. S2R introduces a representation learning framework for multi-view RL and defines a novel MCEB auxiliary objective for the training of the actor and critic to extract useful features from pixel states by ignoring task-irrelevant information. As a decoupling representation module, S2R is easy to integrate with the deep RL agents to find optimal policies. To evaluate S2R, we perform extensive experiments on the DMControl suite. Empirical results show that S2R learns robust representations and improves sample efficiency of the RL agent on various default and noisy visual continuous control tasks.

We want to emphasize that one way to theoretically analyze the sample efficiency of the S2R method is using the sample complexity trait. According to Kakade (2003) and Strehl et al. (2006), the sample complexity of an RL algorithm can be expressed as the amount of experience the RL agent takes to learn to behave well. As an open and challenging problem, theoretical analysis of the sample complexity of the S2R method combined with specified RL algorithms is a clear direction for future work. Besides, a natural extension of S2R is to combine it with model-based planning, which may further improve its sample efficiency. It is well-known that model-based RL algorithms are generally more sample-efficient than model-free RL algorithms. Therefore, for future research, we are interested in incorporating S2R into model-based RL algorithms, first learning an accurate environment model by reducing the model bias, then planning actions through the learned model. Also, integrating S2R with exploration mechanisms is a reasonable way to improve its sample efficiency in RL sparse-reward visual settings. In RL realistic applications, the sparse-reward problem is common and inevitable, and the agent may need to learn policies in environments with sparse or deceptive rewards. Such learning challenges urge us to improve the exploration efficiency of the S2R method in the future.

## Acknowledgements

This work was supported by the National Natural Science Foundation of China (No. 91948303).

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
