# OpenReview forum: "Self-Supervised Representations for Multi-View Reinforcement Learning"
_auai.org/UAI/2022/Conference — UAI 2022 Poster_

### Official Review · Reviewer_sojG · 2022-04-12

**Q2(1) Originality/Novelty:** 3
**Q2(2) Significance/Impact:** 3
**Q2(3) Correctness/Technical Quality:** 3
**Q2(6) Clarity Of Writing:** 3
**Q6 Overall Score:** 6
**Q8 Confidence In Your Score:** 3

**Q1 Summary And Contributions:**

To solve the sample efficiency problem of the the end to end learning mode in DRL, this paper proposed a self-supervision representation method for multi-view RL. Specifically, the multi-view state representation is learned by the conditional entropy bottleneck. Then, the proposed S2R is implemented in SAC to fulfill the RL task. Finally, the effectiveness of the proposed method is demonstrated on the DMControl Suite with three settings.

**Q2 Assessment Of The Paper:**

More detailed information regarding each of these aspects is given below:

**Q2(4) Quality Of Experiments (Optional):**

3: Good: The experimental evaluation is adequate, and the results convincingly support the main claims.

**Q2(5) Reproducibility:**

3: Good: Key resources (e.g., proofs, code, data) are available and key details (e.g., proofs, experimental setup) are sufficiently well-described for competent researchers to confidently reproduce the main results.

**Q3 Main Strengths:**

The contributions of this paper is import for the sample efficiency problem in DRL, which learn the state representation in the self-supervised way, and discard the end to end learning mode in DRL. Also, the proposed method can achieve the multi-view task and the learned state representation is task-relevant.

The paper is well organized and well-written.

The experimental results are convincing, which support the main claim of this paper.

**Q4 Main Weakness:**

There are some points are required to be clarified more clearly, such as the notation and the definition of “view”.

The major weakness of this paper would be the experiments, which are all conducted on benchmarks. It is expected to see the performance on more complex task.

**Q5 Detailed Comments To The Authors:**

1.What is the definition of “multi-view” in your technical and experimental part?
If I understood correctly, the two view in the method of two-view CEB refers to two consecutive states o_1 and o_2, am I right? If so, I am afraid the concept of view is not consistent with the meaning of multi-view in your introduction part.

2.Notation confusion:
In Section 3, s refers to the state, but s in Fig.1 and Section 4 means the latent representation of observations. It is better to make the notations more clearly clarified.

3.Experimental results:

Why don’t you provide the whole learning process curve of the compared methods, but just the results in 100K step and 5000K step in Table 1 are provided? The learning process can say more.

It is better to make the ablation study result in last row of Fig.7 a separate figure.

My major concern for the experiments is the setting of “view”. If the two views just refers to the two consecutive observation, then what is the functionality of the multi-view predictor.
It is better to explicitly clarify this point at the current setting.

**Q7 Justification For Your Score:**

The paper is well written and organized. The technical contribution is novel. The empirical results well support the proposed method.
I just have one concern for the definition of the view.
If my concern point could be solved technically and experimentally, it would be a good paper.

**Q9 Complying With Reviewing Instructions:**

1: Yes.

---

### Official Review · Reviewer_eabg · 2022-04-14

**Q2(1) Originality/Novelty:** 3
**Q2(2) Significance/Impact:** 3
**Q2(3) Correctness/Technical Quality:** 3
**Q2(6) Clarity Of Writing:** 3
**Q6 Overall Score:** 6
**Q8 Confidence In Your Score:** 2

**Q1 Summary And Contributions:**

This paper proposes a self-supervised method for image-based RL. If my understanding is correct, the method obtains multi-view images through data augmentation and uses the conditional entropy bottleneck to train latent states for each view’s image and their fusions. The supervision signal is the corresponding latent states and rewards in the next time step. The authors conduct experiments on DMControl baselines and achieved STOA results.

**Q2 Assessment Of The Paper:**

More detailed information regarding each of these aspects is given below:

**Q2(4) Quality Of Experiments (Optional):**

3: Good: The experimental evaluation is adequate, and the results convincingly support the main claims.

**Q2(5) Reproducibility:**

3: Good: Key resources (e.g., proofs, code, data) are available and key details (e.g., proofs, experimental setup) are sufficiently well-described for competent researchers to confidently reproduce the main results.

**Q3 Main Strengths:**

- The experiment results are strong. The proposed method outperformed all baselines including state-based RL; very data efficient.
- The idea looks promising to me. it seems to take the advantage of both contrastive learning and variational inference.

**Q4 Main Weakness:**

- I did not find the baseline experiments or ablation study that compares the number of views.
- presentation is not very clear; ‘multi-view’ is a little confusing.

**Q5 Detailed Comments To The Authors:**

After reading the paper, I am still confused about what a “view” means here. In computer view, view usually means a view from different view points, meaning the change of the camera position. However, it seems to refer a view the observation after kind of data augmentation. I do not know if my understanding is correct. Either the paper lacks a clear presentation, or the paper uses an unconventional notation. I hope the author can clarify this.

Besides, I do not see the ablation study on number of views used in the paper. Since the paper claims to be multi-view, the number of views should matter. But I do not find such an experiment to answer the question. The single view version of the paper should be a very reasonable baseline to compare. Otherwise it’s hard to justify the necessities of multi-view RL.

Overall I think the proposed method makes a lot of sense to me. If my understanding is correct, it combines the information bottleneck and multi-view inputs, which looks similar to the data augmentation to guide the representation learning. The experiment results are surprising to me as it outperforms the state-based RL on all tasks.

**Q7 Justification For Your Score:**

The paper has pretty good experiment results, outperforms previous SOTA on challenging image-based RL tasks. The idea is novel and I do not find obvious flaw. The writing could be improved, but I didn't find any obvious flaw.

**Q9 Complying With Reviewing Instructions:**

1: Yes.

---

### Official Review · Reviewer_6RUB · 2022-04-23

**Q2(1) Originality/Novelty:** 2
**Q2(2) Significance/Impact:** 2
**Q2(3) Correctness/Technical Quality:** 2
**Q2(6) Clarity Of Writing:** 3
**Q6 Overall Score:** 5
**Q8 Confidence In Your Score:** 4

**Q1 Summary And Contributions:**

This paper presents Self-Supervision Representations (S2R) for multi-view reinforcement learning. The paper shows that S2R can significantly improve the sample efficiency of RL training methods. The paper covers a detailed explanation of how to derive S2R by using a Conditional Entropy Bottleneck (CEB) principle. Based on S2R, the paper propose an auxiliary objective to common model-free RL methods. Experiments results demonstrate the performance of S2R in a DMControl environment.

**Q2 Assessment Of The Paper:**

More detailed information regarding each of these aspects is given below:

**Q2(4) Quality Of Experiments (Optional):**

3: Good: The experimental evaluation is adequate, and the results convincingly support the main claims.

**Q2(5) Reproducibility:**

2: Fair: Key resources (e.g., proofs, code, data) are unavailable but key details (e.g., proof sketches, experimental setup) are sufficiently well-described for an expert to confidently reproduce the main results.

**Q3 Main Strengths:**

1. The paper is easy to follow. The main figures are well-presented. The structure is clear.
2. The empirical results are promising. It seems S2R significantly outperforms other methods. The leading is big. The experiment can generally demonstrate the main arguments of the paper.
3. The S2R objective generally makes sense. Applying CEB to multi-view RL is interesting.
4. The main baselines of S2R are covered and compared.
5 The audiences of this paper match well with the UAI community.

**Q4 Main Weakness:**

1. It is unclear how S2R overcomes the partial observability in POMDP. (see 1 in Q5)
2. The results in the sample efficiency experiments contradict some common beliefs. (see 2 in Q5)
3. It could be difficult for S2R to scale to multiple views. (see 3 in Q5)
4. This paper covers only the model-free methods (SAC). (see 4 in Q5)
5. The experiment setting misses some important details. (see 5 in Q5)
6. Paper presentation could be improved. (see 6 in Q5)
7. No theoretical Insights. (see 7 in Q5)

**Q5 Detailed Comments To The Authors:**

1. **Partial Observability**: This paper assumes a POMDP setting. However, S2R directly extracts states (s) from the current observation (o) (see Page 4, lower left). This design contradicts the definition of POMDP (e.g., see page 3, O(o|s)). I am not sure how S2R overcomes partial observability. Could you explain it? Previous methods (e.g., PlaNet) commonly apply hidden states to store the history information, but it seems S2R skip this step.

2. **Sample Efficiency**: The sample efficiency experiments claim S2R works better than using ground-truth states. I could not understand why latent values could be more efficient than these states in terms of training agents, especially when the latent variables are meaningless at the beginning of training. I believe the sample efficiency of state-SAC should be the upper bound of this experiment, but it is clearly not what we observed. Could you explain it?

3. **Scaling to Multi-View**: I am not sure if the objective in equation (15) can scale well to multiple views: The overlapped expectations will be difficult to approximate. I believe the sample complexity will grow exponentially instead of linearly as N increases.

4. **Model-Free v.s., Model-Based**: S2R predicts the transition dynamics in the environment. Given the next state is predicted, model-based planning could be implemented, but it seems S2R is built with SAC throughout the paper. I could not follow the intuition of this design. In fact, model-based planning can significantly **improve sample efficiency**, which is the main goal of this paper.

5. **Experiments Design**: I read the experiment setup, but I could not understand why the DMControl suite is considered to be multi-view. I might have missed some important details, please let me know where shall I find them.

6. **Presentation**: The paper is generally well-written as I mentioned, but the size of labels in the plots is too small for the readers like me. Equation (10) is difficult to follow. I can not understand what each term means. The presentation could be more friendly if Equation (10) could be presented in the format of meaningful terms like entropy, cross-entropy, and conditional entropy.

7. **Theoretical Insight** It seems this paper has very limited theoretical insight. All the results are presented empirically. I am curious about how the sample complexity grows as N increases, and how the MCEB can smooth the growth rate.


**Q7 Justification For Your Score:**

The paper has some issues in the current format. I am open to discussion and will like to increase my score if my main concerns are well responsed.

**Q9 Complying With Reviewing Instructions:**

1: Yes.

---

### Decision · Program_Chairs · 2022-05-15

**Decision:**

Accept (Poster)

**Comment:**

Meta Review: The paper proposes a new self-supervised learning technique to improve the sample efficiency of RL via data augmentation.  This is a nice contribution that advances the state of the art.  The reviewers raised several points that were clarified by the authors in the rebuttal.  The authors are encouraged to follow the reviewers' suggestion when preparing the final version.